# Industrial Symbiosis Systems: Promoting Carbon Emission Reduction Activities

**DOI:** 10.3390/ijerph16071093

**Published:** 2019-03-27

**Authors:** Haiyan Shan, Junliang Yang, Guo Wei

**Affiliations:** 1China Institute of Manufacturing Development, Nanjing University of Information Science and Technology, Nanjing, Jiangsu 210044, China; 2School of Management Science and Engineering, Nanjing University of Information Science & Technology, Nanjing 210044, China; yangjl@nuist.edu.cn; 3Mathematics & Computer Science, University of North Carolina at Pembroke, Pembroke, NC 28372, USA; guo.wei@uncp.edu

**Keywords:** carbon emission reduction, industrial symbiosis system, tripartite evolutionary game, evolutionary stable strategy

## Abstract

The carbon emission problem in China needs to be solved urgently. Industrial symbiosis, as an effective means to improve resource efficiency, can better alleviate the carbon emission problem. Under such a circumstance, this paper regards an industrial symbiosis system as a collection of producers, consumers and decomposers, and analyzes the strategic selections and behavioral characteristics of their carbon emission reduction activities through a tripartite evolutionary game model, and then the effects of related parameters on the evolutionary stable strategies of stakeholders are discussed. The results demonstrate that: (1) the regular return and the rate of return determine the ability of stakeholders to undertake carbon reduction activities; (2) the initial willingness of stakeholders to participate will affect the evolutionary speed of the strategies; (3) a high opportunity cost reduces the inertia of stakeholders to carry out carbon emission reductions; (4) producers, consumers and decomposers can avoid “free rides” by signing agreements or adopting punitive measures.

## 1. Introduction

With the rapid development of the economy, substantial industrialization progress and fast urbanization, a series of problems caused by carbon emissions, such as global warming leading to extreme climate change and species reduction, have become a heated topic in society. To deal with haze, acid rain or other environmental problems, considering human sustainable development, the government of China made a commitment at the Copenhagen Climate Change Conference that the carbon dioxide emissions per unit of GDP would be reduced by 40%–45% by 2020 [1]. To achieve this goal, China has introduced various policies to promote energy conservation and emission reductions, such as limiting the development of energy-consuming industries and accelerating the transformation of energy-saving technologies. However, the status quo of the current carbon emissions is still grim, and overall carbon emissions are now projected to increase until 2030 before leveling off [2].

### 1.1. Traditional Study Methods

It is not difficult to find by sorting through the past literature that in addition to the purely theoretical analysis made by individual literatures [3,4], most studies first carry out theoretical modeling, then conduct verification or prediction analysis. To some extent, these studies describe a complete carbon reduction decision-making process, and mainly from the following aspects: 

(1) What are the main factors promoting carbon reduction? The economic development levels of different regions are diverse, and the corresponding influential factors may vary, such as energy consumption [5], industrial scale [6], and living standard [7]. In addition, Wu et al. [8] find that industrial structure, energy intensity, population size, and per capita GDP play a significant role in four regions classified by the economic level and carbon intensity, respectively. Li et al. [9] reveal that the development strategies of manufacturing structural rationalization and upgrading that aim to reduce carbon dioxide emissions depend on the level of a region’s resource dependence and industrialization in China. Thus, when the main sources of carbon emissions are analyzed based on the characteristics of regional economic development, the conclusion drawn will be more practical.

(2) How to estimate the regional or industrial emission reduction potential? The previous literature mainly involves non-parametric metafrontier approaches [10], DEA (data envelopment analysis) window analysis approaches [11], and undesirable slack-based measure models [12]. Although the conclusions are different due to the different models used, these results still have certain reference significance for the measurement of emission reduction potential. Furthermore, the potential of emission reduction can better answer the question of whether China’s 2020 emission reduction target can be achieved [13].

(3) How to prevent carbon leakage? Most researchers use the computable general equilibrium model to calculate carbon leakage. Then, different measures are proposed according to the causes of carbon leakage, for instance, carbon tax level adjustment [14], full border adjustment [15], free allowance allocation [16] and so on. After the decision is completed, it is necessary to evaluate the performance of carbon emission reduction projects to judge whether the decision is effective or not. Other methods are also used, such as AIM/Enduse model [17], Monte Carlo simulation [18], tactical planning model [19], etc.

### 1.2. The Perspective of Industrial Symbiosis

Since Anton de Bary, a German biologist, established the theory of symbiosis, some scholars have found similar phenomena in industrial systems [20]. At present, there is no consensus about the connotation of industrial symbiosis (IS). The first concept of IS was proposed by Chertow [21], that is, “IS engages traditionally separate industries in a collective approach to competitive advantage involving physical exchange of materials, energy, water, and/or by-products”. Other concepts of IS have also been defined by scholars according to their research objects or dimensions [22,23,24,25]. Although the definitions are different, they still have similarities which are mainly embodied in three aspects: (1) Make cooperation a prerequisite; (2) Make the exchange of material and energy an approach; (3) Make win-win a result. Compared to the harm from traditional economic development and energy structures, industrial symbiosis shows its superior environmental benefits. Much literature has analyzed the superiority of IS in many different regions, including Kawasaki [26], Lubei [27], Gujiao [28], Oahu [29] and South Korea [30]. Golev and Corder [31] also proposed that the mode of IS can improve the eco-efficiency. Other researchers on industrial symbiosis have mainly focused on structure [32,33,34], evaluation [35,36,37,38], regulation [39,40,41] and evolution [42,43,44,45].

As an important carbon reduction strategy, some studies specifically analyzed the detailed changes of carbon emissions in IS. For instance, Dong et al. [46] employed a hybrid LCA model to evaluate the lifecycle carbon footprint with and without IS, and their results showed that the carbon emission efficiency can be improved with the implementation of IS. Dong et al. [47] reviewed the low-carbon city practice in China, conducted a case study to calculate the CO_2_ emissions reduction potential under promoting IS projects in two cities of China, and quantified the related environmental benefits in an IS network. Kim et al. [48] proposed a practical approach to quantify total and allocate GHG emissions from IS exchanges by integrating the GHG protocols and life cycle assessments. Yu et al. [49] presented a set of methodological rules for the quantification of CO_2_ abatement from IS, and concluded three methods for quantifying CO_2_ emission reduction under different data conditions. Yu et al. [50] applied scenario analysis to examine the effects of IS performance on carbon emission reduction within the Xinfa Group, and found that carbon emissions decreased by 10.84% compared with that under no IS condition. Zhang et al. [51] constructed an optimization model with the constraints of both product demand and carbon emission reduction cost, and carried out a scenario simulation of carbon emission reduction cooperation from perspective of IS for the iron and steel industry.

### 1.3. The Problem

Even though there have been many studies on the carbon emissions issue of industrial symbiosis, and previous studies have offered us plentiful results, another two sides ought to be further discussed which are also the differences between our paper and previous studies.

Firstly, the research perspective. The emphasis of the previous studies is the macro industrial symbiosis system. However, our paper focuses on the components of the industrial symbiosis system, and explores the carbon emission reduction behavior characteristics of different stakeholders and the evolutionary behaviors among them.

Secondly, the methodology. In the past, econometric models or scenario simulations have been used as the main approaches. In this paper, the evolutionary game model is taken as the main method. Different from the traditional game model, it ponders the bounded rationality of the stakeholders [52], and also involves the process of making or optimizing the decision, i.e., the importance and irreversibility of time.

Last but not least, most literatures on industrial symbiosis are limited to the scale of industrial parks, and only selected a particular location as the research object. That means a comprehensive study of the common industrial symbiosis is absent. In contrast, this research emphasizes the industrial symbiosis system itself, which has more general practicability. As a consequence, in this paper, the producers, consumers and decomposers carrying out carbon emission reduction in order to reduce the ecological environment problems constitute the interdependent complex system, to be called industrial symbiosis system. We will explore the carbon emission reductions of these stakeholders based on an evolutionary game model, and analyze the strategy selection for each object in the system.

This paper concentrates on the carbon emissions issue of industrial symbiosis system and makes the following three contributions: (1) modeling the game behavior and evolutionary stable strategy of multiple stakeholders in the industrial symbiosis system through a tripartite evolutionary game model; (2) obtaining the stable evolutionary strategies and the stability conditions under eight kinds of scenarios by solving the model; (3) identifying the factors influencing the stable evolutionary strategies and analyzing the mechanisms of these factors.

The rest of the paper is organized as follows: Section 2 establishes a tripartite evolutionary model of carbon emission reduction in an industrial symbiosis system and puts forward a payoff matrix for the stakeholders. Section 3 explores the stakeholders’ stable evolutionary strategies by analyzing the asymptotic stability of the equilibrium points. Meanwhile, the numerical simulation results of the effects of the parameter variations on the strategies are also presented. Section 4 discusses the results. The conclusions are summarized in Section 5.

## 2. Model

This paper considers an industrial symbiosis system which is composed of multiple producers, consumers and decomposers undertaking the role of energy cycling to reduce their ecological environmental problems. Every time, we select one from each of these three groups and carry out the game of carbon emission reduction (CER) randomly. The hypotheses for the industrial symbiosis system are as follows:
(1)Assume that the sizes of the three groups remain relatively stable. Thus, the scale of each group is standardized to 1. Suppose at time t, the probabilities of CER activities in producer, consumer and decomposer groups are x(t), y(t) and z(t), and hence the probabilities of not to carry out carbon emission reduction (NCER) are 1−x(t), 1−y(t) and 1−z(t) respectively.(2)Each stakeholder has a bounded rationality, so it is not easy to make at optimal strategy at any time. In order to reach a stable state, the strategy will be adjusted and improved constantly.(3)In the industrial symbiosis system, all stakeholders pursue a high-level synergy of economic value and environmental value. For the sake of discussion, here we reflect it in the form of fitness.(4)Each group interacts with the other two groups in the industrial symbiosis system. Specifically:(i)If the stakeholder involved in the abovementioned game carries out CER activity, it will obtain a basic return regardless of whether other stakeholders carry out CER activities or not. Here *f_i_*(*i* = 1, 2, 3) denotes the basic returns for the producer, consumer or decomposer carrying out CER activity, respectively.(ii)If no more than two stakeholders (one or two) are involved in CER activities, the two or one who does not carry out such activity will share the benefits of others, i.e., “free riding” exists. Here, *π_i_*(*i* = 1, 2, 3) indicates the free riding returns of each group, respectively.(iii)The number of stakeholders involved in the activities may affect the fitness. All stakeholders involved in CER activities may not get less benefits than when only one or two are involved in such activities, i.e., *α*_1_ ≥ *α*_0_, *β*_2_ ≥ *β*_1_ ≥ *β*_0_ and *γ*_2_ ≥ *γ*_1_ ≥ *γ*_0_ , where *α_j_*, *β_j_* and *γ_j_* denote the return growth of producer, consumer and decomposer’s respectively when *j* + 1 members are participating in CER (*j* = 0, 1, 2).(5)The stakeholders have to pay some costs while improving their own fitness by CER activities. Suppose *C_i_*(*i* = 1, 2, 3) represent the input costs of the three groups, respectively.

According to the hypotheses mentioned above, the payoff matrix of the tripartite evolutionary game in the industrial symbiosis system can be established, as shown in Table 1.

## 3. Tripartite Evolutionary Model Analysis

### 3.1. Replicated Dynamic Equation

For the producer group, the expected fitness of conducting carbon emission reduction activities is [53]:$$\begin{array}{ll} U_{1y} & =yz((1+\alpha_{2})f_{1}-C_{1})+y(1-z)((1+\alpha_{1})f_{1}-C_{1})\\ & +(1-y)z((1+\alpha_{1})f_{1}-C_{1})+(1-y)(1-z)((1+\alpha_{0})f_{1}-C_{1}), \end{array}$$
that is:
$$U_{1y}=(yz\alpha_{2}+(y(1-z)+(1-y)z)\alpha_{1}+(1-y)(1-z)\alpha_{0})f_{1}+f_{1}-C_{1}.$$

The expected fitness of not conducting emission reduction activities is:$$U_{1n}=yz\pi_{1}+y(1-z)\pi_{1}+(1-y)z\pi_{1}+(1-y)(1-z)f_{1},$$
that is:
$$U_{1n}=(1-(1-y)(1-z))\pi_{1}+(1-y)(1-z)f_{1}.$$

The average expected fitness is:$${\bar{U}}_{1}=xU_{1y}+(1-x)U_{1n}.$$

The replicated dynamic equation of CER activities for the producer group is:$$\begin{array}{lll} \frac{dx}{dt} & = & x(U_{1y}-{\bar{U}}_{1})=x(1-x)(U_{1y}-U_{1n})\\ & = & x(1-x)((yz\alpha_{2}+(y(1-z)+(1-y)z)\alpha_{1}+(1-y)(1-z)\alpha_{0})f_{1}\\ & & +(1-(1-y)(1-z))(f_{1}-\pi_{1})-C_{1}). \end{array}$$

Put:$$\begin{array}{ll} \varphi (y,\;z)= & (yz\alpha_{2}+(y(1-z)+(1-y)z)\alpha_{1}+(1-y)(1-z)\alpha_{0})f_{1}\\ & +(1-(1-y)(1-z))(f_{1}-\pi_{1})-C_{1}. \end{array}$$

The above equation for the producer group can be rewritten as:(1)$$\frac{dx}{dt}=x(1-x)\varphi (y,\;z).$$

Similarly, the replicated dynamic equation of CER activities for the consumer group is:$$\begin{array}{ll} \frac{dy}{dt}= & y(1-y)((xz\beta_{2}+(x(1-z)+(1-x)z)\beta_{1}+(1-x)(1-z)\beta_{0})f_{2}\\ & +(1-(1-x)(1-z))(f_{2}-\pi_{2})-C_{2}). \end{array}$$

Put:$$\begin{array}{ll} \phi (x,\;z)= & (xz\beta_{2}+(x(1-z)+(1-x)z)\beta_{1}+(1-x)(1-z)\beta_{0})f_{2}\\ & +(1-(1-x)(1-z))(f_{2}-\pi_{2})-C_{2}. \end{array}$$

The above equation for the consumer group can be rewritten as:(2)$$\frac{dy}{dt}=y(1-y)\phi (x,\;z).$$

The replicated dynamic equation of CER activities for the decomposer group is:$$\begin{array}{ll} \frac{dz}{dt}= & z(1-z)((xy\gamma_{2}+(x(1-y)+(1-x)y)\gamma_{1}+(1-x)(1-y)\gamma_{0})f_{3}\\ & +(1-(1-x)(1-y))(f_{3}-\pi_{3})-C_{3}). \end{array}$$

Put:$$\begin{array}{ll} \psi (x,\;y)= & (xy\gamma_{2}+(x(1-y)+(1-x)y)\gamma_{1}+(1-x)(1-y)\gamma_{0})f_{3}\\ & +(1-(1-x)(1-y))(f_{3}-\pi_{3})-C_{3}. \end{array}$$

The above equation for the decomposer group can be rewritten as:(3)$$\frac{dz}{dt}=z(1-z)\psi (x,\;y).$$

### 3.2. Equilibrium Solution Analysis of the Industrial Symbiosis System

Equations (1), (2) and (3) constitute a tripartite dynamic system. The solution domain Ω of the tripartite dynamic system is [0, 1]×[0, 1]×[0, 1]. The simultaneous equations are used to find the equilibrium solutions of the evolutionary game:(4)$$\left\{ \begin{array}{l} \frac{dx}{dt}=x(1-x)\varphi (y,\;z)=0\\ \frac{dy}{dt}=y(1-y)\phi (x,\;z)=0\\ \frac{dz}{dt}=z(1-z)\psi (x,\;y)=0 \end{array}\right.$$
differential equations (4) can be rewritten as F(x, y, z)=0.

**Proposition** **1.***The industrial symbiosis system has eight equilibrium points in which each group adopts the pure strategy, and they are (0, 0, 0), (1, 0, 0), (0, 1, 0), (0, 0, 1), (1, 1, 0), (1, 0, 1), (0, 1, 1) and (1, 1, 1)*.

The proof can be found in the Appendix A.

**Proposition** **2.***The industrial symbiosis system has six equilibrium points in which only a single group adopts the pure strategy, and they are: $\left(0,\;\frac{c_{3}-\gamma_{0}f_{3}}{(1+\gamma_{1}-\gamma_{0})f_{3}-\pi_{3}},\;\frac{c_{2}-\beta_{0}f_{2}}{(1+\beta_{1}-\beta_{0})f_{2}-\pi_{2}}\right)$, $\left(\frac{c_{3}-\gamma_{0}f_{3}}{(1+\gamma_{1}-\gamma_{0})f_{3}-\pi_{3}},\;0,\;\frac{c_{1}-\alpha_{0}f_{1}}{(1+\alpha_{1}-\alpha_{0})f_{1}-\pi_{1}}\right)$, $\left(\frac{c_{2}-\beta_{0}f_{2}}{(1+\beta_{1}-\beta_{0})f_{2}-\pi_{2}},\;\frac{c_{1}-\alpha_{0}f_{1}}{(1+\alpha_{1}-\alpha_{0})f_{1}-\pi_{1}},\;0\right)$, $\left(1,\;\frac{c_{3}+\pi_{3}-(1+\gamma_{1})f_{3}}{(\gamma_{2}-\gamma_{1})f_{3}},\;\frac{c_{2}+\pi_{2}-(1+\beta_{1})f_{2}}{(\beta_{2}-\beta_{1})f_{2}}\right)$, $\left(\frac{c_{3}+\pi_{3}-(1+\gamma_{1})f_{3}}{(\gamma_{2}-\gamma_{1})f_{3}},\;1,\;\frac{c_{1}+\pi_{1}-(1+\alpha_{1})f_{1}}{(\alpha_{2}-\alpha_{1})f_{1}}\right)$ and $\left(\frac{c_{2}+\pi_{2}-(1+\beta_{1})f_{2}}{(\beta_{2}-\beta_{1})f_{2}},\;\frac{c_{1}+\pi_{1}-(1+\alpha_{1})f_{1}}{(\alpha_{2}-\alpha_{1})f_{1}},\;1\right)$. The existence conditions of the first equilibrium point are $0<\frac{c_{3}-\gamma_{0}f_{3}}{(1+\gamma_{1}-\gamma_{0})f_{3}-\pi_{3}}<1$ and $0<\frac{c_{2}-\beta_{0}f_{2}}{(1+\beta_{1}-\beta_{0})f_{2}-\pi_{2}}<1$, and the existence conditions of other equilibrium points can be obtained similarly*.

The proof can be found in the Appendix A.

**Proposition** **3.***The industrial symbiosis system has 12 equilibrium points in which two groups adopt the pure strategy. The specific equilibrium points and the conditions are shown in Table 2*.

The proof can be found in the Appendix A.

In fact, the conditions given by Proposition 3 can also be observed from the payoff matrix of the industrial symbiosis system. For example, when the consumer group carries out CER activities and the decomposer group does not carry out CER activities, if the fitness of carrying out CER by producers is equal to that of no CER activities, i.e., $(1+\alpha_{1})f_{1}-c_{1}=\pi_{1}$, the producer group chooses a mixed strategy due to the fact that they do not have any preferences for either of these two strategies.

**Remark** **1.***When the related parameters of carbon emission reduction for producer, consumer and decomposer groups satisfy certain conditions, a mixed strategy equilibrium point $\left(x^{*},\;y^{*},\;z^{*}\right)$ may exist in the industrial symbiosis system and $x^{*},\;y^{*},\;z^{*}\in (0,\;1)$*.

For example, if $\alpha_{0}=\beta_{0}=\gamma_{0}=\frac{1}{8}$, $\alpha_{1}=\beta_{1}=\gamma_{1}=\frac{1}{4}$, $\alpha_{2}=\beta_{2}=\gamma_{2}=\frac{3}{8}$, $f_{i}=\frac{5}{4}$, $C_{i}=\frac{1}{2}$ and $\pi_{i}=1\;(i=1,\;2,\;3)$, it can be checked that $\left(\frac{1}{2},\;\frac{1}{2},\;\frac{1}{2}\right)$ is a mixed equilibrium point of the system.

### 3.3. Stability Analysis of Industrial Symbiosis System

The probabilities of CER activities of producer, consumer and decomposer groups may vary over time. During the period, the evolutionary stable strategy (ESS) is more likely to attract our attention. However, the equilibrium point obtained by the replicated dynamic equation is not necessarily the ESS of the system. According to the theory of stability proposed by Lyapunov [54], the asymptotic stability of the system at the equilibrium point can be judged through analyzing the eigenvalues of the Jacobian matrix of the system, i.e., the necessary and sufficient condition for the asymptotic stability of the system is that all the eigenvalues of the Jacobian matrix have a negative real part. Then, we get the partial derivatives of x, y and z in turn for Equations (4). The final Jacobian matrix can be shown as follows:$$J=\left(\begin{array}{ccc} \frac{\partial F_{1}}{\partial x} & \frac{\partial F_{1}}{\partial y} & \frac{\partial F_{1}}{\partial z}\\ \frac{\partial F_{2}}{\partial x} & \frac{\partial F_{2}}{\partial y} & \frac{\partial F_{2}}{\partial z}\\ \frac{\partial F_{3}}{\partial x} & \frac{\partial F_{3}}{\partial y} & \frac{\partial F_{3}}{\partial z} \end{array}\right)=\left(\begin{array}{ccc} F_{11} & F_{12} & F_{13}\\ F_{21} & F_{22} & F_{23}\\ F_{31} & F_{32} & F_{33} \end{array}\right)$$
where:
$F_{11}=(1-2x)\varphi (y,z)$,$F_{12}=x(1-x)((z\alpha_{2}+(1-2z)\alpha_{1}-(1-z)\alpha_{0})f_{1}+(1-z)(f_{1}-\pi_{1}))$,$F_{13}=x(1-x)((y\alpha_{2}+(1-2y)\alpha_{1}-(1-y)\alpha_{0})f_{1}+(1-y)(f_{1}-\pi_{1}))$,$F_{21}=y(1-y)((z\beta_{2}+(1-2z)\beta_{1}-(1-z)\beta_{0})f_{2}+(1-z)(f_{2}-\pi_{2}))$,$F_{22}=(1-2y)\phi (x,z)$,$F_{23}=y(1-y)((x\beta_{2}+(1-2x)\beta_{1}-(1-x)\beta_{0})f_{2}+(1-x)(f_{2}-\pi_{2}))$,$F_{31}=z(1-z)((y\gamma_{2}+(1-2y)\gamma_{1}-(1-y)\gamma_{0})f_{3}+(1-y)(f_{3}-\pi_{3}))$,$F_{32}=z(1-z)((x\gamma_{2}+(1-2x)\gamma_{1}-(1-x)\gamma_{0})f_{3}+(1-x)(f_{3}-\pi_{3}))$,$F_{33}=(1-2z)\psi (x,y)$.

The characteristic equation of matrix *J* can be solved to analyze the asymptotic stability of the industrial symbiosis system.

#### 3.3.1. All Groups Adopt the Pure Strategy

Based on the eight pure strategy equilibrium points of the industrial symbiosis system (see Proposition 1 in Section 3.2), the conditions of ESS are summarized in Table 3 when all groups adopt the pure strategy. The proof of Table 3 is provided in Appendix A.

#### 3.3.2. Single Group Adopts Pure Strategy

Proposition 2 provides six general expressions when only one group adopts the pure strategy in the industrial symbiosis system. Proposition 4 shows that all of the above equilibrium solutions are not the evolutionary stable strategies.

**Proposition** **4.***The six equilibrium points when only one single group adopts the pure strategy in the industrial symbiosis system, i.e.: $\left(0,\;\frac{c_{3}-\gamma_{0}f_{3}}{(1+\gamma_{1}-\gamma_{0})f_{3}-\pi_{3}},\;\frac{c_{2}-\beta_{0}f_{2}}{(1+\beta_{1}-\beta_{0})f_{2}-\pi_{2}}\right)$, $\left(\frac{c_{3}-\gamma_{0}f_{3}}{(1+\gamma_{1}-\gamma_{0})f_{3}-\pi_{3}},\;0,\;\frac{c_{1}-\alpha_{0}f_{1}}{(1+\alpha_{1}-\alpha_{0})f_{1}-\pi_{1}}\right)$, $\left(\frac{c_{2}-\beta_{0}f_{2}}{(1+\beta_{1}-\beta_{0})f_{2}-\pi_{2}},\;\frac{c_{1}-\alpha_{0}f_{1}}{(1+\alpha_{1}-\alpha_{0})f_{1}-\pi_{1}},\;0\right)$, $\left(1,\;\frac{c_{3}+\pi_{3}-(1+\gamma_{1})f_{3}}{(\gamma_{2}-\gamma_{1})f_{3}},\;\frac{c_{2}+\pi_{2}-(1+\beta_{1})f_{2}}{(\beta_{2}-\beta_{1})f_{2}}\right)$, $\left(\frac{c_{3}+\pi_{3}-(1+\gamma_{1})f_{3}}{(\gamma_{2}-\gamma_{1})f_{3}},\;1,\;\frac{c_{1}+\pi_{1}-(1+\alpha_{1})f_{1}}{(\alpha_{2}-\alpha_{1})f_{1}}\right)$, $\left(\frac{c_{2}+\pi_{2}-(1+\beta_{1})f_{2}}{(\beta_{2}-\beta_{1})f_{2}},\;\frac{c_{1}+\pi_{1}-(1+\alpha_{1})f_{1}}{(\alpha_{2}-\alpha_{1})f_{1}},\;1\right)$ are not the evolutionary stable strategies*.

The proof can be found in the Appendix A.

#### 3.3.3. Other Situations

Proposition 3 puts forward 12 equilibrium points in which two groups adopt the pure strategy. In this case, each equilibrium point in the system represents some possible states of one certain group. Thus, it is impossible to analyze its evolutionary stability. Therefore, the evolutionary stable strategy of two groups adopting the pure strategy is no longer analyzed.

Remark 1 illustrates that the industrial symbiosis system may have a mixed strategy equilibrium point $\left(x^{*},\;y^{*},\;z^{*}\right)$ when the related parameters of three groups satisfy some conditions. Considering the general expression involves many parameters and the computation is complex, the exact solution cannot be given here. However, once the mixed strategy is obtained, the conditions of ESS can also be analyzed.

**Remark** **2.***If the related parameters given meet the equilibrium conditions of the mixed strategy, then the conditions of ESS for the mixed strategy can be written out in accordance with the theory of Jacobian matrix and Cardano formula*.

The deductive process is provided in the Appendix A.

### 3.4. Numerical Simulations

Due to the symmetry of our model, we choose $E_{1}$, $E_{2}$, $E_{5}$ and $E_{8}$ for numerical simulations to explore the evolutionary process of each stakeholder, and the correctness of the game model is verified according to the evolutionary stable results.

Suppose $\alpha_{0}=\beta_{0}=\gamma_{0}=0.1$, $\alpha_{1}=\beta_{1}=\gamma_{1}=0.2$, $\alpha_{2}=\beta_{2}=\gamma_{2}=0.3$, $f_{i}=4$, $\pi_{i}=5.5$ and $C_{i}=0.5\;(i=1,\;2,\;3)$, and we get the equilibrium point $E_{1}(0,\;0,\;0)$ which is the unique evolutionary stable strategy when $\alpha_{0}f_{1}<c_{1}$, $\beta_{0}f_{2}<c_{2}$ and $\gamma_{0}f_{3}<c_{3}$. According to the simulation results (see Figure 1a), regardless of the initial states of the stakeholders, the probabilities of CER for each group will continually decrease as the evolution goes on. The added value of profits of the stakeholders in carbon emission reduction is less than their input costs which means they lack incentives of carbon emission reduction. Finally, the optimal strategies of them is “NCER”, and the evolution stable strategy of the tripartite evolutionary game is $E_{1}(0,\;0,\;0)$.

Suppose $\alpha_{0}=\beta_{0}=\gamma_{0}=0.1$, $\alpha_{1}=\beta_{1}=\gamma_{1}=0.2$, $\alpha_{2}=\beta_{2}=\gamma_{2}=0.3$, $f_{1}=6$, $f_{2}=f_{3}=4$, $\pi_{i}=5.5$ and $C_{i}=0.5\;(i=1,\;2,\;3)$, and we get the equilibrium point $E_{2}(1,\;0,\;0)$ which is the unique evolutionary stable strategy when $\alpha_{0}f_{1}>c_{1}$, $(\beta_{1}+1)f_{2}-c_{2}<\pi_{2}$ and $(\gamma_{1}+1)f_{3}-c_{3}<\pi_{3}$. From Figure 1b, the probabilities of the consumers and decomposers will continually decrease, whereas the probabilities of the producers in carbon emission reduction constantly increase. The evolutionary stable strategy will converge to $E_{2}(1,\;0,\;0)$ in the end. It is clear that the producers are more willing to reduce the carbon emission reduction when the added return of carbon emission reduction is larger than the input cost. At this time, “free ride” will occur in the industrial symbiosis system, once the added values from stakeholders’ carbon emission reduction are less than the free riding returns. Therefore, they are more willing to choose “NCER” as their optimal strategy.

Suppose $\alpha_{0}=\beta_{0}=\gamma_{0}=0.1$, $\alpha_{1}=\beta_{1}=\gamma_{1}=0.2$, $\alpha_{2}=\beta_{2}=\gamma_{2}=0.3$, $f_{1}=f_{2}=6$, $f_{3}=4$, $\pi_{i}=5.5$ and $C_{i}=0.5\;(i=1,\;2,\;3)$, and we derive the equilibrium point $E_{5}(1,\;1,\;0)$ in Figure 1b, which is the unique evolutionary stable strategy when $(\alpha_{1}+1)f_{1}-c_{1}>\pi_{1}$, $(\beta_{1}+1)f_{2}-c_{2}>\pi_{2}$ and $(\gamma_{2}+1)f_{3}-c_{3}<\pi_{3}$. For any initial point, the added values from the producers and consumers’ carbon emission reduction are higher than the free riding returns, thus the probabilities of them in carbon emission reduction will continually increase. On the other hand, the decomposer group’s added value is less than its free riding return, thus it will choose “NCER” as the optimal strategy. Eventually, the evolutionary stable strategy is $E_{5}(1,\;1,\;0)$ in the tripartite evolutionary game.

Finally, Suppose $\alpha_{0}=\beta_{0}=\gamma_{0}=0.1$, $\alpha_{1}=\beta_{1}=\gamma_{1}=0.2$, $\alpha_{2}=\beta_{2}=\gamma_{2}=0.3$, $f_{i}=6$, $\pi_{i}=5.5$ and $C_{i}=0.5\;(i=1,\;2,\;3)$, and we derive the equilibrium point $E_{8}(1,\;1,\;1)$ which is the unique evolutionary stable strategy when $(\alpha_{2}+1)f_{1}-c_{1}>\pi_{1}$, $(\beta_{2}+1)f_{2}-c_{2}>\pi_{2}$ and $(\gamma_{2}+1)f_{3}-c_{3}>\pi_{3}$. According to the results from Figure 1d, the probabilities of all stakeholders carrying out emission reduction will constantly increase to 1. Since all added values of carbon emission reduction are higher than the “free riding” option, “CER” is their best stable choice. Consequently, the stable evolutionary strategy is $E_{8}(1,\;1,\;1)$.

### 3.5. Parameter Analysis

Thus, in the following we analyze the effects of several parameters on the evolutionary results of the probabilities that each group carries out emission reduction activities.

#### 3.5.1. The Initial State

In Figure 2, the evolutionary results under the different initial states of the industrial symbiosis system are given. The initial states of the first batch are x(0)=0.1, y(0)=0.2 and z(0)=0.4, and the initial states of the second batch are x(0)=0.15, y(0)=0.3 and z(0)=0.5. The other parameters are $\alpha_{0}=\beta_{0}=\gamma_{0}=0.1$, $\alpha_{1}=\beta_{1}=\gamma_{1}=0.2$, $\alpha_{2}=\beta_{2}=\gamma_{2}=0.3$, $f_{i}=5$, $\pi_{i}=5.5$ and $C_{i}=0.5\;(i=1,\;2,\;3)$.

As shown in Figure 2, at the beginning, the growth rates of emission reduction probability are slower; with time going on, the growth rates are much faster; later the growth rates all slowly converge to carry out emission reduction activities. The growth rate of emission reduction probability is similar to the logistic curve. Two sets of data all converge to carry out emission reduction activities, but the convergence rates are different: they can quickly converge to the stable solution if the willingness is much larger in the initial state; conversely, the convergence speed is much slower.

In fact, if the initial states are changed but other parameters remain unchanged, the industrial symbiosis system still can converge to all carry out emission reduction activities as long as the related parameters satisfy the ESS condition 8 in Table 3.

#### 3.5.2. The Input Cost of Emission Reduction

In Figure 3, the evolutionary results of the probabilities of emission reduction activities for each group are given under different input costs of emission reduction in the industrial symbiosis systems. The input costs of emission reduction of the first batch for each group are all equal to 0.5, and the input costs of the second batch are 0.5, 0.5 and 0.25. The other parameters are x(0)=0.15, y(0)=0.3, z(0)=0.5, $\alpha_{0}=\beta_{0}=\gamma_{0}=0.1$, $\alpha_{1}=\beta_{1}=\gamma_{1}=0.2$, $\alpha_{2}=\beta_{2}=\gamma_{2}=0.3$, $f_{i}=4$ and $\pi_{i}=5.5\;(i=1,\;2,\;3)$.

As shown in Figure 3, when the decomposer’s input cost of emission reduction is reduced, its emission reduction probability decreases slowly in the initial stage but then quickly rises and gradually converges to 1. That is to say, the probability of the stakeholders to carry out the emission reduction activities increases gradually with the reduction of the input costs of the emission reduction and even converges to 1. In fact, if the input cost of emission reduction is decreased, its adaptability will be increased which will lead to the increase of the willingness to reduce emissions.

#### 3.5.3. The Free Riding Return of Emission Reduction

In Figure 4, the evolutionary results of the probabilities of emission reduction activities for each group are given under different free riding returns of the industrial symbiosis systems. The free riding returns of the first batch for each group are all equal to 5.5, and the free riding returns of the second batch are 5.5, 5.5 and 5.8. The other parameters are x(0)=0.15, y(0)=0.3, z(0)=0.5, $\alpha_{0}=\beta_{0}=\gamma_{0}=0.1$, $\alpha_{1}=\beta_{1}=\gamma_{1}=0.2$, $\alpha_{2}=\beta_{2}=\gamma_{2}=0.3$, $C_{i}=0.5$ and $f_{i}=5\;(i=1,\;2,\;3)$.

As Figure 4 shows, with the decomposer’s free riding return increasing, its willingness in CER will decrease greatly and gradually converge to 0. Contrarily, it will rise sharply, and even gradually converge to 1. Although there is no change between the producer and consumer’s free riding returns, the decreasing decomposer’s willingness in emission reduction generates a result that the probabilities of the producer and consumer’s emission reduction no longer rise significantly. We can conclude that the increase of the free riding return will reduce the willingness for emission reduction, so that the probability of carrying out emission reduction activities falls and even converges to zero.

#### 3.5.4. The Basic Return and the Rate of Return of Emission Reduction Input

In Figure 5, the evolutionary results of the probabilities of emission reduction activities for each group are given under different basic returns of the industrial symbiosis systems. The basic returns of the first and second batch are $f_{i}=4$ and $f_{i}=5\;(i=1,\;2,\;3)$, respectively. The other parameters are x(0)=0.15, y(0)=0.3, z(0)=0.5, $\alpha_{0}=\beta_{0}=\gamma_{0}=0.1$, $\alpha_{1}=\beta_{1}=\gamma_{1}=0.2$, $\alpha_{2}=\beta_{2}=\gamma_{2}=0.3$, $C_{i}=0.5$ and $\pi_{i}=5.5\;(i=1,\;2,\;3)$.

From Figure 5, the greater the basic return, the higher the probability of the emission reduction activity. Particularly, if the relevant parameters satisfy the condition 8 in Table 3, all stakeholders will converge to carry out emission reduction activities. Otherwise, the probability of the emission reduction activity will decrease, even converges to no emission reduction activity. Its impact on the probability of emission reduction activity is mainly due to the fitness which is positively related to the basic return regardless of whether other stakeholders carry out CER activities or not.

In Figure 6, the evolutionary results of the probabilities of emission reduction activities for each group are given under different rate of return of emission reduction input in the industrial symbiosis system. The specific values of the two sets of the parameters are shown in Figure 6 and other parameters are x(0)=0.15, y(0)=0.3, z(0)=0.5, $f_{i}=5$, $\pi_{i}=5.5$ and $C_{i}=0.5\;(i=1,\;2,\;3)$. Figure 6 illustrates that with a decreasing rate of return, the willingness of each stakeholder in CER will decrease significantly, and even gradually converge to 0. Contrarily, it will rise sharply, and even gradually converges to 1.

## 4. Discussion

With the rapid economic development, China has become the largest carbon emitter in the world [55], and CER is bound to become an important factor in the management of enterprises. As society pays more attention to environmental protection issues, the preference for low-carbon products has become increasingly obvious [56]. However, traditional enterprises in China generally have the characteristics of high energy consumption and high emissions [57]. Therefore, they have the desire to reduce carbon emissions, but this will generate high costs which will lead to the difficulty of making decisions on CER.

Industrial symbiosis is a proven strategy to limit carbon emissions whilst increasing resource-efficiency and business growth [58]. Through the cooperation on emission reduction between enterprises in the industrial symbiosis system, the carbon metabolism efficiency of the system can be improved, and more carbon dioxide emissions can be avoided [59]. Such cooperation is an important way for enterprises to develop a low carbon economy.

However, in this process, the external positive effect of any enterprise emission reduction provides others with an incentive of “free riding”. It is the particularity of the symbiosis system and the uncertainty of decision-making between enterprises that motivates us to analyze the CER activities in the industrial symbiosis system from the perspective of game theory between the related groups.

### 4.1. Implications

From our theoretical analysis of the tripartite evolutionary game model, we find that the stakeholders will participate and stabilize in CER if the difference between the benefit and the input cost of emission reduction is higher than the benefit of free ride, such as the producers and consumers in $E_{5}$, else they may as well enjoy the benefits of free ride, such as the decomposers in $E_{5}$. In the absence of the free riding returns, the stakeholders will participate in CER when the added benefit is higher than its cost of emission reduction such as the producers in $E_{2}$, else they have no incentive to carry out the CER activities such as the consumers and decomposers in $E_{2}$.

The numerical simulations demonstrate that some parameters play an important role on determining the evolutionary stable strategy in industrial symbiosis system:

(1) When the input cost of emission reduction is high, the stakeholders are not likely to reduce emissions. As a result, the government needs to introduce some subsidies or rewards to enhance the willingness of enterprises to actively invest in emission reduction or innovation. For example, the State Council of China issued the State Council’s Opinions on the Development of Overcapacity in the Iron and Steel Industry in 2016. In the third part of the Opinions, Article 8 “Policy Measures” points out that the Chinese government should strengthen the support of rewards and subsidies, which is conducive to stimulating the enthusiasm and initiative of enterprises in energy saving and emission reduction. The Ministry of Finance has also formulated the Interim Measures for the Management of Subsidies for Energy Conservation and Emission Reduction.

(2) The higher the free riding return, the lower the willingness to participate in CER, and vice versa. All stakeholders in the industrial symbiosis system can interact with each other. Therefore, when one conducts CER activities, the other can benefit from it, called the free riding return. High free riding return is not a favorable phenomenon for an industrial symbiosis system. Olson laid out a theory of collective inaction [60]. What might be of mutual benefit is not achieved. If many individuals decide to “free-ride” on the actions of others, the others may stop contributing to the collective good. In order to avoid the negative effects of free ride, the stakeholders can build some contracts to bind each other. Dongguan Eco-Park can draw lessons from this conclusion. Being located at the edge of several towns, the polluted water has trans-regional mobility, so it is difficult to know which township enterprises discharge secretly, which seriously hinders the development of ecological parks.

(3) The higher the basic return, the greater willingness to participate in emission reduction. Actually, some companies have high regular returns, however, they sometimes are reluctant to reduce carbon emission for their products or services. Yunnan Luoping Zinc and Electricity Co in China is an example, which is the first joint-stock enterprise integrating hydroelectric power generation, mining, zinc smelting and deep processing. Since 2015, the company has been punished by the local environmental protection bureau and environmental protection supervision group repeatedly, and its annual net profit was more than 17 million RMB, but it has not yet taken any substantial rectification action until severely notified by the official website of the Ministry of Ecological Environment in 2018.

(4) The higher the rate of return of emission reduction input, the greater the possibility of the stakeholders to reduce carbon emissions, such as high energy consumption and high polluting industry. In recent years, China’s iron and steel industry has been vigorously promoting the strategy of capacity reduction, it has ushered in development opportunities. Taking Baosteel in China as an example, its annual profit in 2016 was 9 billion RMB, while in 2017 was 19.2 billion RMB. In fact, Baosteel reduced its excess capacity by 11 million tons during this two-year period. The company states that “it adheres to the path of innovation, coordination, green, open and shared development in order to achieve the vision of realizing the sustainable development of green iron and steel industry ecosphere, and creating the competitive green brand [61]”. A brief empirical research on the Baosteel is provided in the Appendix B.

(5) The ultimate evolutionary stable strategy may remain unchanged when other parameters been fixed while the initial state is altered. Even if the initial willingness of each group to reduce carbon emission is low, the stakeholders will all gradually converge to carry out carbon emission. The initial willingness is not associated with the conditions of ESS, however it can affect the convergence speed. This means that the initial awareness of the decision-makers plays a vital role in the decision-making process: the higher the initial willingness of decision maker to reduce emission, the faster the speed of reaching stability. Meanwhile, it reveals the necessity to strengthen the concept of corporate carbon reduction education.

### 4.2. Our Model vs. Other Evolutionary Game Models

Existing research on stability analysis of actual systems mainly focuses on bilateral evolutionary game models. Recently, tripartite evolutionary game models have penetrated into various fields and become an important means for analyzing behavioral science. For example, Zhang et al. [62], Sheng and Webber [63], and Liu et al. [64] used the tripartite evolutionary game models to study waste cooking oil-to-energy, south-to-north water diversion, and coal-mine safety regulation, respectively. However, their analysis of the game models still has some limitations or defects, as detailed below.

Zhang et al. [62] only considered the scenario that all groups adopt pure strategies, but ignored the scenario of mixed strategies. Meanwhile, the numerical simulation of the game model is missing. Sheng and Webber [63] calculated the scenario in which all groups simultaneously adopt mixed strategies, but no further analysis was given. They also did not consider the scenario in which only one group or two groups select mixed strategies. According to Liu et al. [64], it is noteworthy that their model consists of one individual in group A, and two individuals in group B, i.e., only two kinds of stakeholders are involved in their model. Similarly, the mixed strategy scenario is also not considered.

It is not difficult to find that all the above studies neglect the scenarios in which stakeholders adopt mixed strategies in the game. From the perspective of completeness, it is not reasonable and may lead to the omission of equilibrium solutions. Thus, in order to overcome these limitations or defects, we conduct the corresponding theorems and deduction processes of the mixed strategies of one group, two groups and three groups in the tripartite evolutionary game model.

## 5. Conclusions

Some environmental problems such as gray haze phenomenon and water pollution are essentially the response of resource inefficiency. A symbiosis activity among industries based on by-products can save the resources and protect the environment through a closed material cycle and the cascaded utilization of energy.

This study focuses on the bounded rational CER activities of the producer, consumer and decomposer groups in the industrial symbiosis system, constructs the evolutionary game model and gives the equilibrium points that one group, two groups or three groups adopt pure strategy or mixed strategy. We also analyze the evolutionary stable strategies and explore the effects of relevant parameters on the evolutionary results of the probability of emission reduction activities for all stakeholders. The main conclusions are as follows:

(1) There are eight equilibrium points that all groups adopt the pure strategy, and if every stakeholder’s difference between the return and cost is higher than the benefit of free ride, they will be stable in carrying out CER activities.

(2) The industrial symbiosis system will be stable in the state that every group adopts the pure strategy once the system satisfies certain conditions. However, the industrial symbiosis system will not be stable in the state that only a single group adopts the pure strategy.

(3) Some factors play a vital role on the evolutionary results of the probability of emission reduction activities, such as the initial state, the input cost of emission reduction, the free riding return of emission reduction, the basic return and the rate of return of emission reduction input.

Specifically, if the initial probability of the stakeholder’s emission reduction activities is larger, it can converge to the stable solution much faster; conversely, the convergence rate is slower. The willingness of carbon emission reduction is positively related to the basic return and the rate of return of emission reduction input, and negatively related to the input cost of emission reduction and the free riding return of emission reduction.

The model analyzed in this paper does not consider the role of government, such as supervision over the enterprises. Accordingly, such model is applicable to self-organized industrial symbiosis systems and is not appropriate to hetero-organized systems which are formed by external instructions or planned by government in advance. In future, we will consider the government supervision acts in CER. This pattern not only involves competition among enterprises, but also includes competition between enterprises and government. Thus, a more complicated analysis is required for this pattern which is a challenging work.

## Figures and Tables

**Figure 1 ijerph-16-01093-f001:**
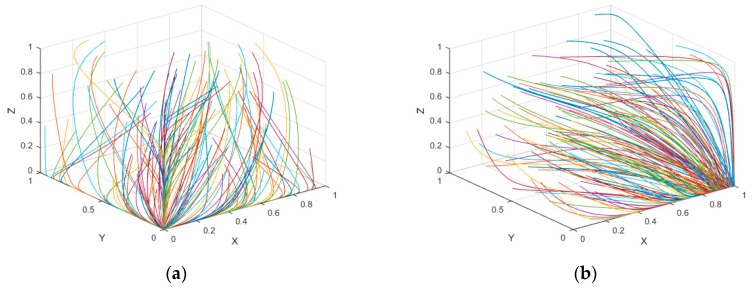
Evolutionary process of each stakeholder. (**a**) $E_{1}(0,\;0,\;0)$ (**b**) $E_{2}(1,\;0,\;0)$ (**c**) $E_{5}(1,\;1,\;0)$ (**d**) $E_{8}(1,\;1,\;1)$.

**Figure 2 ijerph-16-01093-f002:**
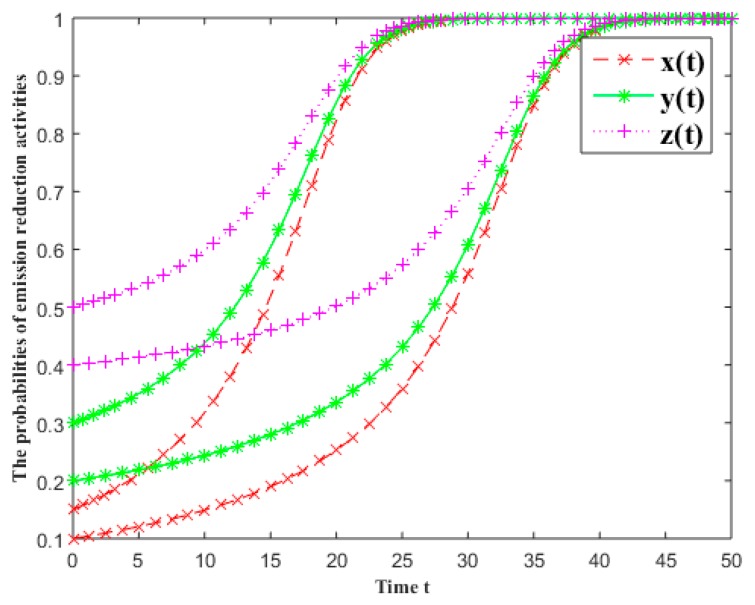
The effects of the initial states on the probabilities of emission reduction activities.

**Figure 3 ijerph-16-01093-f003:**
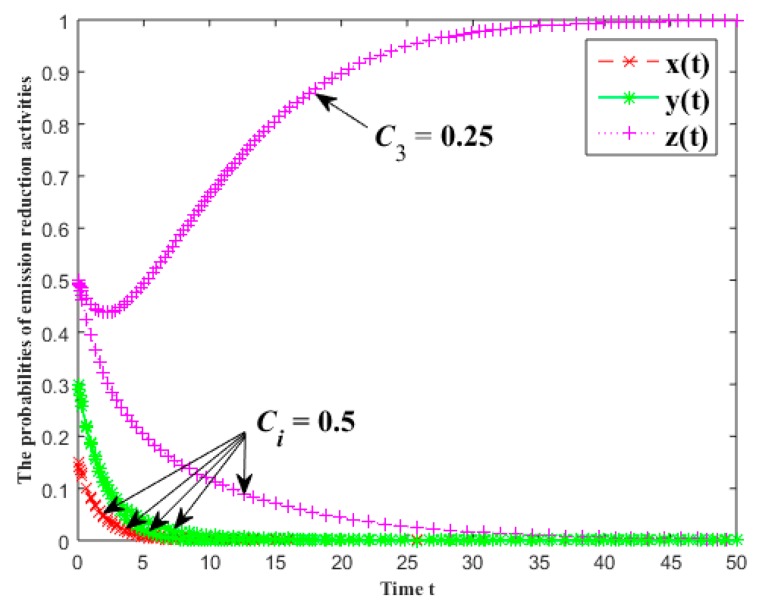
The effects of the input costs of emission reduction on the probabilities of emission reduction activities.

**Figure 4 ijerph-16-01093-f004:**
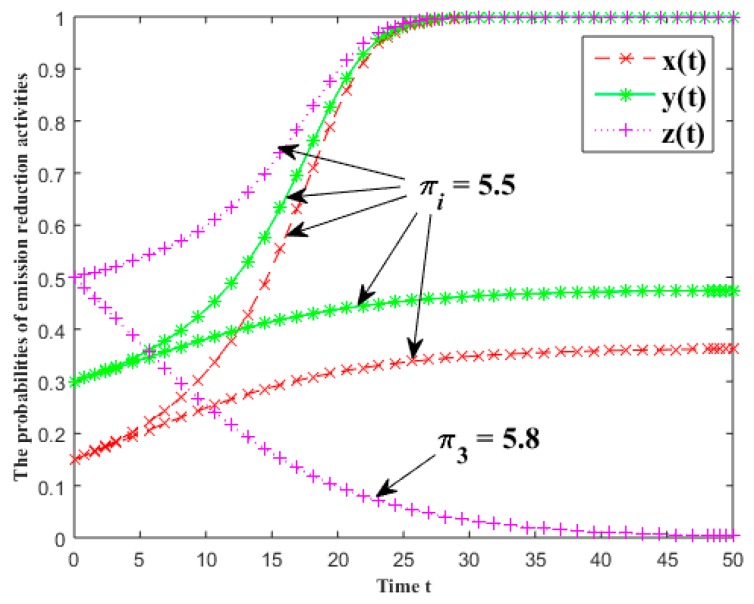
The effects of the free riding return of emission reduction on the probabilities of emission reduction activities.

**Figure 5 ijerph-16-01093-f005:**
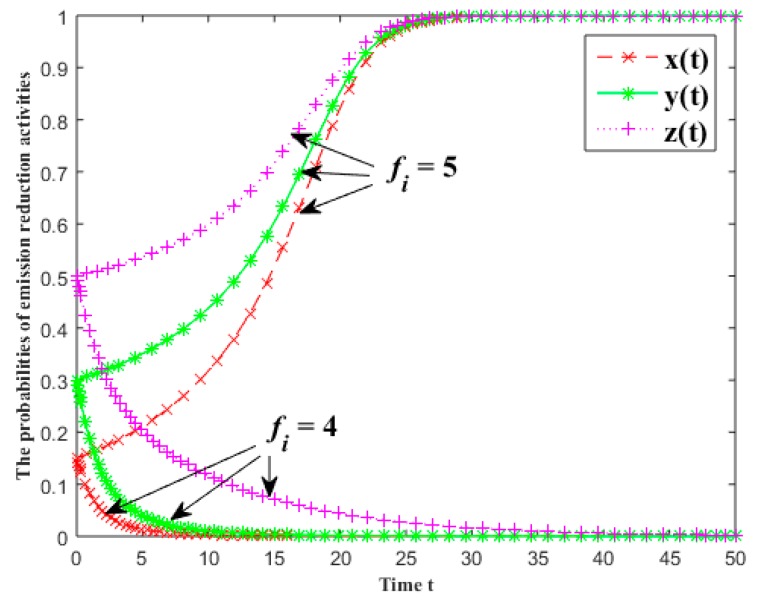
The effects of the basic returns on the probabilities of emission reduction activities.

**Figure 6 ijerph-16-01093-f006:**
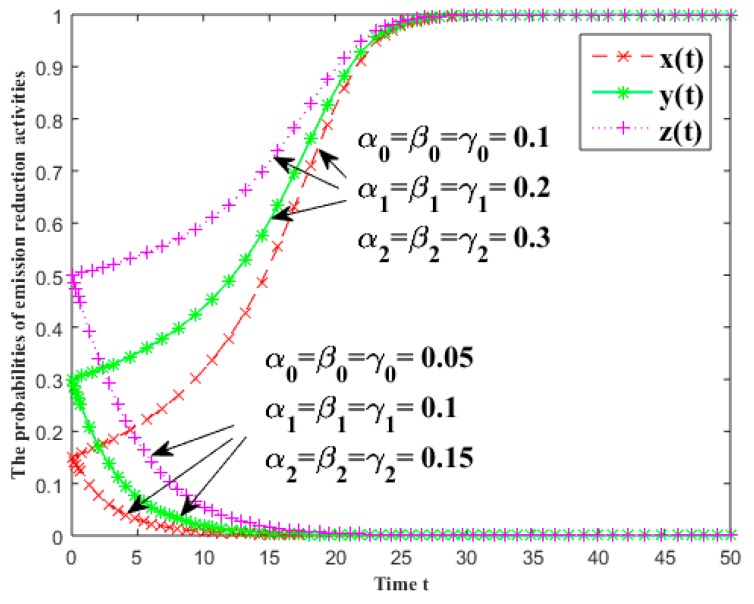
The effects of the rate of return of emission reduction input on the probabilities of emission reduction activities.

**Table 1 ijerph-16-01093-t001:** The payoff matrix of the tripartite evolutionary game.

				Decomposer	
				CER (z)	NCER (1−z)
**Producer**	CER (x)	**Consumer**	CER (y)	$(1+\alpha_{2})f_{1}-C_{1}$, $(1+\beta_{2})f_{2}-C_{2}$, $(1+\gamma_{2})f_{3}-C_{3}$	$(1+\alpha_{1})f_{1}-C_{1}$, $(1+\beta_{1})f_{2}-C_{2}$, $\pi_{3}$
			NCER (1−y)	$(1+\alpha_{1})f_{1}-C_{1}$, $\pi_{2}$, $(1+\gamma_{1})f_{3}-C_{3}$	$(1+\alpha_{0})f_{1}-C_{1}$, $\pi_{2}$, $\pi_{3}$
	NCER (1−x)	**Consumer**	CER (y)	$\pi_{1}$, $(1+\beta_{1})f_{2}-C_{2}$, $(1+\gamma_{1})f_{3}-C_{3}$	$\pi_{1}$, $(1+\beta_{0})f_{2}-C_{2}$, $\pi_{3}$
			NCER (1−y)	$\pi_{1}$, $\pi_{2}$, $(1+\gamma_{0})f_{3}-C_{3}$	$f_{1}$, $f_{2}$, $f_{3}$

**Table 2 ijerph-16-01093-t002:** The conditions of equilibrium point for two groups adopting pure strategy

Mixed Strategy State	Equilibrium Point	Condition of Equilibrium Point
x∈(0, 1)	(x, 0, 0)	$\alpha_{0}f_{1}=c_{1}$
	(x, 1, 0)	$(1+\alpha_{1})f_{1}-c_{1}=\pi_{1}$
	(x, 0, 1)	
	(x, 1, 1)	$(1+\alpha_{2})f_{1}-c_{1}=\pi_{1}$
y∈(0, 1)	(0, y, 0)	$\beta_{0}f_{2}=c_{2}$
	(0, y, 1)	$(1+\beta_{1})f_{2}-c_{2}=\pi_{2}$
	(1, y, 0)	
	(1, y, 1)	$(1+\beta_{2})f_{2}-c_{2}=\pi_{2}$
z∈(0, 1)	(0, 0, z)	$\gamma_{0}f_{3}=c_{3}$
	(0, 1, z)	$(1+\gamma_{1})f_{3}-c_{3}=\pi_{3}$
	(1, 0, z)	
	(1, 1, z)	$(1+\gamma_{2})f_{3}-c_{3}=\pi_{3}$

**Table 3 ijerph-16-01093-t003:** The conditions of ESS for all groups adopting pure strategy.

Case	Equilibrium Point	Conditions of ESS
I	$E_{1}(0,\;0,\;0)$	$\alpha_{0}f_{1}<c_{1}$, $\beta_{0}f_{2}<c_{2}$, $\gamma_{0}f_{3}<c_{3}$
II	$E_{2}(1,\;0,\;0)$	$\alpha_{0}f_{1}>c_{1}$, $(\beta_{1}+1)f_{2}-c_{2}<\pi_{2}$, $(\gamma_{1}+1)f_{3}-c_{3}<\pi_{3}$
	$E_{3}(0,\;1,\;0)$	$(\alpha_{1}+1)f_{1}-c_{1}<\pi_{1}$, $\beta_{0}f_{2}>c_{2}$, $(\gamma_{1}+1)f_{3}-c_{3}<\pi_{3}$
	$E_{4}(0,\;0,\;1)$	$(\alpha_{1}+1)f_{1}-c_{1}<\pi_{1}$, $(\beta_{1}+1)f_{2}-c_{2}<\pi_{2}$, $\gamma_{0}f_{3}>c_{3}$
III	$E_{5}(1,\;1,\;0)$	$(\alpha_{1}+1)f_{1}-c_{1}>\pi_{1}$, $(\beta_{1}+1)f_{2}-c_{2}>\pi_{2}$, $(\gamma_{2}+1)f_{3}-c_{3}<\pi_{3}$
	$E_{6}(1,\;0,\;1)$	$(\alpha_{1}+1)f_{1}-c_{1}>\pi_{1}$, $(\beta_{2}+1)f_{2}-c_{2}<\pi_{2}$, $(\gamma_{1}+1)f_{3}-c_{3}>\pi_{3}$
	$E_{7}(0,\;1,\;1)$	$(\alpha_{2}+1)f_{1}-c_{1}<\pi_{1}$, $(\beta_{1}+1)f_{2}-c_{2}>\pi_{2}$, $(\gamma_{1}+1)f_{3}-c_{3}>\pi_{3}$
IV	$E_{8}(1,\;1,\;1)$	$(\alpha_{2}+1)f_{1}-c_{1}>\pi_{1}$, $(\beta_{2}+1)f_{2}-c_{2}>\pi_{2}$, $(\gamma_{2}+1)f_{3}-c_{3}>\pi_{3}$

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
