# Peer review of "Industrial Symbiosis Systems: Promoting Carbon Emission Reduction Activities"

_ijerph, 2019, doi:10.3390/ijerph16071093_

Round 1
Reviewer 1 Report
The model is described in detail (pages 4 - 13) - has it been applied to any case studies in order to validate it? for example Baosteel and Yunnan Luoping Zinc and Electricity Co are mentioned on page 18.
The 10 pages devoted to listing the components of the model are at present too long, the manuscript would be alot more readable and of interest to readers if this was shortened to the key points and instead illustrated its functioning with application to actual case studies. Alternatively you may want to consider submission of the manuscript to a journal whose scope might be more appropriate for a predominantly theoretical manuscript.
page 18 line 514 'Baosteel in China adhere to the path of innovation, coordination, green, open and shared development in order to achieve the vision of becoming the most competitive steel enterprise in the world.' - is this a quote from Baosteel?
Author Response
Comment 1.
The model is described in detail (pages 4 - 13) - has it been applied to any case studies in order to validate it? For example, Baosteel and Yunnan Luoping Zinc and Electricity Co are mentioned on page 18.
Response 1: Thank you for your advice. Some management implications on the basis of model construction and numerical analysis are analyzed in Section 4.1. The validation in the revised manuscript is presented as follows.
(1) We listed some policies issued by the State Council to illustrate that some subsidies or reward can enhance the willingness of enterprises to actively invest in emission reduction or innovation. These changes are made in Section 4.1, from line 407 to line 413.
(2) Olson’s opinions and the example of Dongguan Eco-Park are applied in Section 4.1, from line 417 to line 424, to show the harm of “free-ride” and the importance of building some contracts.
(3) The examples of Yunnan Luoping Zinc and Electricity Co are particularized from line 427 to line 433 in the revised manuscript to illustrate the situation in which the stakeholders who have higher regular returns, are sometimes, however, reluctant to reduce carbon emission for their products or services.
(4) Baosteel is provided as a concrete example in Section 4.1, line 436 to line 442, to illustrate the finding the higher the rate of return on emissions reductions, the greater the likelihood of reducing carbon emissions by the stakeholders.
Comment 2.
The 10 pages devoted to listing the components of the model are at present too long, the manuscript would be a lot more readable and of interest to readers if this was shortened to the key points and instead illustrated its functioning with application to actual case studies. Alternatively you may want to consider submission of the manuscript to a journal whose scope might be more appropriate for a predominantly theoretical manuscript.
Response 2: We have shortened the components of the model, from 10 pages to 6 pages in the revised manuscript. The proofs of Propositions 1, 2, 3 and 4, proof of Table 3 and the deduction process of Remark 2 are moved to an appendix.
Comment 3.
Page 18 line 514 'Baosteel in China adhere to the path of innovation, coordination, green, open and shared development in order to achieve the vision of becoming the most competitive steel enterprise in the world.' - is this a quote from Baosteel?
Response 3: This is a quote from Baosteel's official website, citation source has been identified in the reference [61].

Reviewer 2 Report
Dear author your paper is very intersting, I really appreciated how his work use a different methodology compared to previous study.
But for my opinion there are some aspects to improve.
In the introduction is necessary to explain better because is important reduce the carbon emission.
In these sense, see and cite.
Li, Z., Shao, S., Shi, X., Sun, Y., & Zhang, X. (2019). Structural transformation of manufacturing, natural resource dependence, and carbon emissions reduction: Evidence of a threshold effect from China. Journal of Cleaner Production, 206, 920-927
Wu, Y., Shen, L., Zhang, Y., Shuai, C., Yan, H., Lou, Y., & Ye, G. (2019). A new panel for analyzing the impact factors on carbon emission: A regional perspective in China. Ecological indicators, 97, 260-268.
The metodology section and the presentation of the results is very clear.
But from linee 200 to 217, i suggest you to insert a table, to exlplain clear this part.
The conclusion is very well written,
Consider the results achieved, highlight the limits of the model used and identify future research fields.
Author Response
Comment 1.
In the introduction is necessary to explain better because is important reduce the carbon emission.
In these sense, see and cite.
Li, Z., Shao, S., Shi, X., Sun, Y., & Zhang, X. (2019). Structural transformation of manufacturing, natural resource dependence, and carbon emissions reduction: Evidence of a threshold effect from China. Journal of Cleaner Production, 206, 920-927.
Wu, Y., Shen, L., Zhang, Y., Shuai, C., Yan, H., Lou, Y., & Ye, G. (2019). A new panel for analyzing the impact factors on carbon emission: A regional perspective in China. Ecological indicators, 97, 260-268.
Response 1: Thank you for providing these references. In Section 1.1, these two papers are added and brief discussions are provided in the revised manuscript (lines 46 to 51).
Comment 2.
The methodology section and the presentation of the results is very clear.
But from line 200 to 217, I suggest you to insert a table, to explain clear this part.
The conclusion is very well written. Consider the results achieved, highlight the limits of the model used and identify future research fields.
Response 2: Following the suggestion, we have inserted a table (Table 2) in Section 3.2 to summarize the main results in Proposition 3.

Round 2
Reviewer 1 Report
Thank you for the revision to the manuscript.
The concern expressed regarding the hypothetical nature of the manuscript in the first set of comments remain. There is no input data from the companies cited as case studies presented in the manuscript on which to base the output of the mathematical models presented.
The quote from Baosteel needs to be in quotation marks, i would also suggest that the text 'It adheres to the path of innovation, coordination, green, open and shared development...' is replaced by 'The company states that it adheres to the path of innovation, coordination, green, open and shared development...'
I would recommend the manuscript is proof read further
Author Response
Comment 1.
The concern expressed regarding the hypothetical nature of the manuscript in the first set of comments remain. There is no input data from the companies cited as case studies presented in the manuscript on which to base the output of the mathematical models presented.
Response 1: Following the suggestion, we take the intra-industrial symbiosis system in Baosteel in China as a case study to validate our results, which is provided in the Appendix B from line 613 to line 645.
Comment 2.
The quote from Baosteel needs to be in quotation marks, I would also suggest that the text 'It adheres to the path of innovation, coordination, green, open and shared development...' is replaced by 'The company states that it adheres to the path of innovation, coordination, green, open and shared development...'
Response 2: We have made quotation marks and replacement from line 438 to line 441.
Comment 3.
I would recommend the manuscript is proof read further.
Response 3: We have checked the English language.
Thank you for your suggestions.

Round 3
Reviewer 1 Report
The additional detail added as Appendix B has improved the manuscript.